# DOUBLE Q-LEARNING: NEW ANALYSIS AND SHARPER FINITE-TIME BOUND

## ABSTRACT

Double Q-learning (Hasselt, 2010) has gained significant success in practice due to its effectiveness in overcoming the overestimation issue of Q-learning. However, theoretical understanding of double Q-learning is rather limited and the only existing finite-time analysis was recently established in Xiong et al. (2020) under a polynomial learning rate. This paper analyzes the more challenging case with a rescaled linear/constant learning rate for which the previous method does not appear to be applicable. We develop new analytical tools that achieve an order-level better finite-time convergence rate than the previously established result. Specifically, we show that synchronous double Q-learning attains an $\epsilon$-accurate global optimum with a time complexity of $\Omega\left(\frac{\ln D}{(1-\gamma)^7\epsilon^2}\right)$, and the asynchronous algorithm attains a time complexity of $\tilde{\Omega}\left(\frac{L}{(1-\gamma)^7\epsilon^2}\right)$, where $D$ is the cardinality of the state-action space, $\gamma$ is the discount factor, and $L$ is a parameter related to the sampling strategy for asynchronous double Q-learning. These results improve the order-level dependence of the convergence rate on all major parameters $(\epsilon, 1-\gamma, D, L)$ provided in Xiong et al. (2020). The new analysis in this paper presents a more direct and succinct approach for characterizing the finite-time convergence rate of double Q-learning.

## 1 INTRODUCTION

Double Q-learning proposed in Hasselt (2010) is a widely used model-free reinforcement learning (RL) algorithm in practice for searching for an optimal policy (Zhang et al., 2018a;b; Hessel et al., 2018). Compared to the vanilla Q-learning proposed in Watkins & Dayan (1992), double Q-learning uses two Q-estimators with their roles randomly selected at each iteration, respectively for estimating the maximum Q-function value and updating the Q-function. In this way, the overestimation of the action-value function in vanilla Q-learning can be effectively mitigated, especially when the reward is random or prone to errors (Hasselt, 2010; Hasselt et al., 2016; Xiong et al., 2020). Moreover, double Q-learning has been shown to have the desired performance in both finite state-action setting (Hasselt, 2010) and infinite setting (Hasselt et al., 2016) where it successfully improved the performance of deep Q-network (DQN), and thus inspired many variants (Zhang et al., 2017; Abed-alguni & Ottom, 2018) subsequently.

In parallel to its empirical success in practice, the theoretical convergence properties of double Q-learning has also been explored. Its asymptotic convergence was first established in Hasselt (2010). The asymptotic mean-square error for double Q-learning was studied in Weng et al. (2020c) under the assumption that the algorithm converges to a unique optimal policy. Furthermore, in Xiong et al. (2020), the finite-time convergence rate has been established for double Q-learning with a polynomial learning rate $\alpha = 1/t^\omega, \omega \in (0,1)$. Under such a choice for the learning rate, they showed that double Q-learning attains an $\epsilon$-accurate optimal Q-function at a time complexity approaching to but never reaching $\Omega(\frac{1}{\epsilon^2})$ at the cost of an asymptotically large exponent on $\frac{1}{1-\gamma}$. However, a polynomial learning rate typically does not offer the best possible convergence rate, as having been shown for RL algorithms that a so-called *rescaled linear learning rate* (with a form of $\alpha_t = \frac{a}{b+ct}$) and a *constant learning rate* achieve a better convergence rate (Bhandari et al., 2018; Wainwright, 2019a;b; Chen et al., 2020; Qu & Wierman, 2020). Therefore, a natural question arises as follows:

*Can a rescaled linear learning rate or a constant learning rate improve the convergence rate of double Q-learning order-wisely? If yes, does it also improve the dependence of the convergence rate on other important parameters of the Markov decision process (MDP) such as the discount factor and the cardinality of the state and action spaces?*

The answer to the above question does not follow immediately from Xiong et al. (2020), because the finite-time analysis framework in Xiong et al. (2020) does not handle such learning rates to yield a desirable result. This paper develops a novel analysis approach and provides affirmative answers to the above question.

## 1.1 OUR CONTRIBUTIONS

This paper establishes sharper finite-time bounds for double Q-learning with a rescaled linear/constant learning rate, which are orderwisely better than the existing bounds in Xiong et al. (2020). We devise a different analysis approach from that in Xiong et al. (2020), which is more capable of handling variants of double Q-learning.

- For synchronous double Q-learning, where all state-action pairs are visited at each iteration, we apply a rescaled linear learning rate $\alpha_t = \frac{3}{3+(1-\gamma)t}$ and show that the algorithm can attain an $\epsilon$-accurate global optimum with a time complexity of $\Omega\left(\frac{\ln D}{(1-\gamma)^7\epsilon^2}\right)$, where $\gamma$ is the discount factor and $D = |\mathcal{S}||\mathcal{A}|$ is the cardinality of the finite state-action space. As a comparison, for the $\epsilon$ dominated regime (with relatively small $\gamma$), our result attains an $\epsilon$-accurate optimal Q-function with a time complexity $\Omega(\frac{1}{\epsilon^2})$, whereas the result in Xiong et al. (2020) (see Table 1) does not exactly reach $\Omega(\frac{1}{\epsilon^2})$ and its approaching to such an order ($\eta := 1 - \omega \to 0$) is at an additional cost of an asymptotically large exponent on $\frac{1}{1-\gamma}$. For $1 - \gamma$ dominated regime, our result improves on that in Xiong et al. (2020) (which has been optimized in the dependence on $1 - \gamma$ in Table 1) by $\mathcal{O}\left(\left(\ln \frac{1}{1-\gamma}\right)^7\right)$.

- For asynchronous double Q-learning, where only one state-action pair is visited at each iteration, we obtain a time complexity of $\tilde{\Omega}\left(\frac{L}{(1-\gamma)^7\epsilon^2}\right)$, where $L$ is a parameter related to the sampling strategy in Assumption 1. As illustrated in Table 1, our result improves upon that in Xiong et al. (2020) order-wisely in terms of its dependence on $\epsilon$ and $1 - \gamma$ as well as on $L$ by at least $\mathcal{O}\left(L^5\right)$.

Our analysis takes a different approach from that in Xiong et al. (2020) in order to handle the rescaled linear/constant learning rate. More specifically, to deal with a pair of nested stochastic approximation (SA) recursions, we directly establish the dependence bound of the error dynamics (of the outer SA) between the Q-estimator and the global optimum on the error propagation (of the inner SA) between the two Q-estimators. Then we develop a bound on the inner SA, integrate it into that on the outer SA as a noise term, and establish the final convergence bound. This is a very different yet more direct approach than that in Xiong et al. (2020), the latter of which captures the blockwise convergence by constructing two complicated block-wisely decreasing bounds for the two SAs. The sharpness of the bound also requires careful selection of the rescaled learning rates and proper usage of their properties.

## 1.2 RELATED WORK

**Theory on double Q-learning:** Double Q-learning was proposed and proved to converge asymptotically in Hasselt (2010). In Weng et al. (2020c), the authors explored the properties of mean-square errors for double Q-learning both in the tabular case and with linear function approximation, under the assumption that a unique optimal policy exists and the algorithm can converge. The most relevant work to this paper is Xiong et al. (2020), which established the first finite-time convergence rate for tabular double Q-learning with a polynomial learning rate. This paper provides sharper finite-time convergence bounds for double Q-learning, which requires a different analysis approach.

**Tabular Q-learning and convergence under various learning rates:** Proposed in Watkins & Dayan (1992) under finite state-action space, Q-learning has aroused great interest in its theoretical study. Its asymptotic convergence has been established in Tsitsiklis (1994); Jaakkola et al. (1994);

Table 1: Comparison of time complexity for (a)synchronous double Q-learning.
The choices $\omega \to 1, \omega = \frac{6}{7}$, and $\omega = \frac{2}{3}$ respectively optimize the dependence of time complexity on $\epsilon, 1 - \gamma$, and $L$ in Xiong et al. (2020). We denote $a \vee b = \max\{a, b\}, a \wedge b = \min\{a, b\}$.

| SyncDQ | Stepsize | Time complexity | |
|---|---|---|---|
| Xiong et al. (2020) | $\frac{1}{t^\omega}, \omega \in (\frac{1}{3}, 1)$ | $\omega = 1 - \eta \to 1$ $\tilde{\Omega}\left(\frac{1}{\epsilon^{2+\eta}} \vee \left(\frac{1}{1-\gamma}\right)^{\frac{1}{\eta}}\right)$ | $\omega = 6/7$ $\tilde{\Omega}\left(\frac{1}{(1-\gamma)^7}\left(\frac{1}{\epsilon^{3.5}} \vee \left(\ln \frac{1}{1-\gamma}\right)^7\right)\right)$ |
| This work | $\frac{3}{3+(1-\gamma)t}$ | $\Omega\left(\frac{1}{\epsilon^2}\right)$ | $\Omega\left(\frac{1}{(1-\gamma)^7\epsilon^2}\right)$ |

| AsyncDQ | Stepsize | Time complexity | | |
|---|---|---|---|---|
| Xiong et al. (2020) | $\frac{1}{t^\omega}, \omega \in (\frac{1}{3}, 1)$ | $\omega = 1 - \eta \to 1$ $\tilde{\Omega}\left(\frac{1}{\epsilon^{2+\eta}} \vee \left(\frac{1}{1-\gamma}\right)^{\frac{1}{\eta}}\right)$ | $\omega = 6/7$ $\tilde{\Omega}\left(\frac{1}{(1-\gamma)^7}\left(\frac{1}{\epsilon^{3.5}} \vee \left(\ln \frac{1}{1-\gamma}\right)^7\right)\right)$ | $\omega = 2/3$ $\tilde{\Omega}\left(\frac{L^6(\ln L)^{1.5}}{(1-\gamma)^9\epsilon^3}\right)$ |
| This work | $\epsilon^2(1-\gamma)^6 \wedge 1$ | $\tilde{\Omega}\left(\frac{1}{\epsilon^2}\right)$ | $\tilde{\Omega}\left(\frac{1}{(1-\gamma)^7\epsilon^2}\right)$ | $\tilde{\Omega}\left(\frac{L}{(1-\gamma)^7\epsilon^2}\right)$ |

Borkar & Meyn (2000); Melo (2001); Lee & He (2019) by requiring the learning rates to satisfy $\sum_{t=0}^\infty \alpha_t = \infty$ and $\sum_{t=0}^\infty \alpha_t^2 < \infty$. Another line of research focuses on the finite-time analysis of Q-learning under different choices of the learning rates. Szepesvári (1998) captured the first convergence rate of Q-learning using a linear learning rate (i.e., $\alpha_t = \frac{1}{t}$). Under similar learning rates, Even-Dar & Mansour (2003) provided finite-time results for both synchronous and asynchronous Q-learning with a convergence rate being exponentially slow as a function of $\frac{1}{1-\gamma}$. Another popular choice is the polynomial learning rate which has been studied for synchronous Q-learning in Wainwright (2019b) and for both synchronous/asynchronous Q-learning in Even-Dar & Mansour (2003). With this learning rate, however, the convergence rate still has a gap with the lower bound of $\mathcal{O}(\frac{1}{\sqrt{T}})$ (Azar et al., 2013). To handle this, a more sophisticated rescaled linear learning rate was introduced for synchronous Q-learning (Wainwright, 2019b; Chen et al., 2020) and asynchronous Q-learning (Qu & Wierman, 2020), and thus yields a better convergence rate. The finite-time bounds for Q-learning were also given with constant stepsizes (Beck & Srikant, 2012; Chen et al., 2020; Li et al., 2020). In this paper, we focus on the rescaled linear/constant learning rate and obtain sharper finite-time bounds for double Q-learning.

**Q-learning with function approximation:** When the state-action space is considerably large or even infinite, the Q-function is usually approximated by a class of parameterized functions. In such a case, Q-learning has been shown not to converge in general (Baird, 1995). Strong assumptions are typically needed to establish the convergence of Q-learning with linear function approximation (Bertsekas & Tsitsiklis, 1996; Melo et al., 2008; Zou et al., 2019; Chen et al., 2019; Du et al., 2019; Yang & Wang, 2019; Jia et al., 2019; Weng et al., 2020a;b) or neural network approximation (Cai et al., 2019; Xu & Gu, 2019). The convergence analysis of double Q-learning with function approximation raises new technical challenges and can be an interesting topic for future study.

## 2 PRELIMINARIES ON DOUBLE Q-LEARNING

We consider a Markov decision process (MDP) over a finite state space $\mathcal{S}$ and a finite action space $\mathcal{A}$ with the total cardinality given by $D := |\mathcal{S}||\mathcal{A}|$. The transition kernel of the MDP is given by $\mathbb{P} : \mathcal{S} \times \mathcal{A} \times \mathcal{S} \to [0, 1]$ denoted as $\mathbb{P}(\cdot|s, a)$. We denote the *random* reward function at time $t$ as $R_t : \mathcal{S} \times \mathcal{A} \times \mathcal{S} \mapsto [0, R_{max}]$, with $\mathbb{E}[R_t(s, a, s')] = R_{sa}^{s'}$. A policy $\pi := \pi(\cdot|s)$ captures the conditional probability distribution over the action space given state $s \in \mathcal{S}$. For a policy $\pi$, we define Q-function $Q^\pi \in \mathbb{R}^{|\mathcal{S}| \times |\mathcal{A}|}$ as

$$Q^\pi(s, a) := \mathbb{E}\left[\sum_{t=1}^\infty \gamma^t R_t(s_t, a_t, s_t') \Big| s_1 = s, a_1 = a\right], \tag{1}$$

where $\gamma \in (0, 1)$ is the discount factor, $a_t \sim \pi(\cdot|s_t)$, and $s_t' \sim \mathbb{P}(\cdot|s_t, a_t)$.

Both vanilla Q-learning (Watkins & Dayan, 1992) and double Q-learning (Hasselt, 2010) aim to find the optimal Q-function $Q^*$ which is the unique fixed point of the Bellman operator $\mathcal{T}$ (Bertsekas & Tsitsiklis, 1996) given by

$$\mathcal{T}Q(s,a) = \mathbb{E}_{s' \sim \mathbb{P}(\cdot|s,a)} \left[ R_{sa}^{s'} + \gamma \max_{a' \in \mathcal{A}} Q(s',a') \right]. \tag{2}$$

Note that the Bellman operator $\mathcal{T}$ is $\gamma$-contractive which satisfies $\|\mathcal{T}Q - \mathcal{T}Q'\| \le \gamma \|Q - Q'\|$ under the supremum norm $\|Q\| := \max_{s,a} |Q(s,a)|$.

The idea of double Q-learning is to keep two Q-tables (i.e., Q-function estimators) $Q^A$ and $Q^B$, and randomly choose one Q-table to update at each iteration based on the Bellman operator computed from the other Q-table. We next describe synchronous and asynchronous double Q-learning algorithms in more detail.

**Synchronous double Q-learning:** Let $\{\beta_t\}_{t \ge 1}$ be a sequence of i.i.d. Bernoulli random variables satisfying $\mathbb{P}(\beta_t = 0) = \mathbb{P}(\beta_t = 1) = 0.5$. At each time $t$, $\beta_t = 0$ indicates that $Q^B$ is updated, and otherwise $Q^A$ is updated. The update at time $t \ge 1$ can be written in a compact form as,

$$\begin{cases} Q_{t+1}^A(s,a) = (1 - \alpha_t \beta_t) Q_t^A(s,a) + \alpha_t \beta_t \left( R_t(s,a,s') + \gamma Q_t^B(s',a^*) \right), \\ Q_{t+1}^B(s,a) = (1 - \alpha_t(1 - \beta_t)) Q_t^B(s,a) + \alpha_t(1 - \beta_t) \left( R_t(s,a,s') + \gamma Q_t^A(s',b^*) \right), \end{cases} \tag{3}$$

for all $(s,a) \in \mathcal{S} \times \mathcal{A}$, where $s'$ is sampled independently for each $(s,a)$ by $s' \sim \mathbb{P}(\cdot|s,a)$, $a^* = \arg\max_{a \in \mathcal{A}} Q^A(s',a)$, $b^* = \arg\max_{a \in \mathcal{A}} Q^B(s',a)$ and $\alpha_t$ is the learning rate. Note that the rewards for both updates of $Q_{t+1}^A$ and $Q_{t+1}^B$ are the same copy of $R_t$.

**Asynchronous double Q-learning:** Different from synchronous double Q-learning, at each iteration the asynchronous version samples only one state-action pair to update the chosen Q-estimator. That is, at time $t$, only the chosen Q-estimator and its value at the sampled state-action pair $(s_t, a_t)$ will be updated. We model this by introducing an indicator function $\tau_t(s,a) = \mathbb{1}_{\{(s_t,a_t)=(s,a)\}}$. Then the update at time $t \ge 1$ of asynchronous double Q-learning can be written compactly as

$$\begin{cases} Q_{t+1}^A(s,a) = (1 - \alpha_t \tau_t(s,a)\beta_t) Q_t^A(s,a) + \alpha_t \tau_t(s,a)\beta_t \left( R_t + \gamma Q_t^B(s',a^*) \right), \\ Q_{t+1}^B(s,a) = (1 - \alpha_t \tau_t(s,a)(1 - \beta_t)) Q_t^B(s,a) + \alpha_t \tau_t(s,a)(1 - \beta_t) \left( R_t + \gamma Q_t^A(s',b^*) \right), \end{cases} \tag{4}$$

for all $(s,a) \in \mathcal{S} \times \mathcal{A}$, where $R_t$ is evaluated as $R_t(s,a,s')$.

In the above update rules (3) and (4), at each iteration only one of the two Q-tables is randomly chosen to be updated. This chosen Q-table generates a greedy optimal action, and the other Q-table is used for estimating the corresponding Bellman operator (or evaluating the greedy action) for updating the chosen table. Specifically, if $Q^A$ is chosen to be updated, we use $Q^A$ to obtain the optimal action $a^*$ and then estimate the corresponding Bellman operator using $Q^B$ to update $Q^A$. As shown in Hasselt (2010), $\mathbb{E}[Q^B(s',a^*)]$ is likely smaller than $\mathbb{E}\max_a[Q^A(s',a)]$, where the expectation is taken over the randomness of the reward for the same $(s,a,s')$ tuple. Such a two-estimator framework adopted by double Q-learning can effectively reduce the overestimation.

Without loss of generality, we assume that $Q^A$ and $Q^B$ are initialized with the same value (usually both all-zero tables in practice). For both synchronous and asynchronous double Q-learning, it has been shown in Xiong et al. (2020) that either Q-estimator is uniformly bounded by $\frac{R_{\max}}{1-\gamma}$ throughout the learning process. Specifically, for either $i \in \{A, B\}$, we have $\|Q_t^i\| \le \frac{R_{\max}}{1-\gamma}$ and $\|Q_t^i - Q^*\| \le \frac{2R_{\max}}{1-\gamma} := V_{\max}$ for all $t \ge 1$. This boundedness property will be useful in our finite-time analysis.

## 3 FINITE-TIME CONVERGENCE ANALYSIS

In this section, we start with modeling the error dynamics to be nested SAs, following by a convergence result for a general SA that will be applicable for both SAs. Then we provide the finite-time results for both synchronous and asynchronous double Q-learning. Finally, we sketch the proof of the main theorem for the synchronous algorithm to help understand the technical proofs.

### 3.1 CHARACTERIZATION OF THE ERROR DYNAMICS

In this subsection, we characterize the (a)synchronous double Q-learning algorithms as a pair of nested SA recursions, where the outer SA recursion captures the error dynamics between the Q-estimator and the global optimum $Q^*$, and the inner SA captures the error propagation between the two Q-estimators which enters into the outer SA as a noise term. Such a characterization enjoys useful properties that will facilitate the finite-time analysis.

**Outer SA:** Denote the iteration error by $r_t = Q_t^A - Q^*$ and define the empirical Bellman operator $\widehat{\mathcal{T}}_t Q(s,a) := R_t(s,a,s') + \gamma \max_{a' \in \mathcal{A}} Q(s',a')$. Then we can have for all $t \geq 1$ (see Appendix C),

$$r_{t+1}(s,a) = (1 - \tilde{\alpha}_t(s,a))r_t(s,a) + \tilde{\alpha}_t(s,a)\left(\mathcal{G}_t(r_t)(s,a) + \varepsilon_t(s,a) + \gamma\nu_t(s',a^*)\right), \quad (5)$$

where $\varepsilon_t := \widehat{\mathcal{T}}_t Q^* - Q^*, \nu_t := Q_t^B - Q_t^A, \mathcal{G}_t(r_t) := \widehat{\mathcal{T}}_t Q_t^A - \widehat{\mathcal{T}}_t Q^* = \widehat{\mathcal{T}}_t(r_t + Q^*) - \widehat{\mathcal{T}}_t Q^*$, and the equivalent learning rate $\tilde{\alpha}_t(s,a) := \begin{cases} \alpha_t \beta_t, & \text{for synchronous version} \\ \alpha_t \beta_t \tau_t(s,a), & \text{for asynchronous version} \end{cases}$. Note that it is by design that we use the same sampled reward $R_t$ in both $\widehat{\mathcal{T}}_t Q^*$ and $\widehat{\mathcal{T}}_t Q_t^A$ in the definition of $\mathcal{G}_t(r_t)$.

These newly introduced variables have several important properties. First of all, the noise term $\{\varepsilon_t\}_t$ is a sequence of i.i.d. random variables satisfying $\mathbb{E}\varepsilon_t = \mathbb{E}[\widehat{\mathcal{T}}_t Q^*] - Q^* = \mathcal{T}Q^* - Q^* = \mathbf{0} \in \mathbb{R}^D$. Furthermore, define the span seminorm of $Q^*$ as $\|Q^*\|_{\text{span}} := \max_{(s,a) \in \mathcal{S} \times \mathcal{A}} Q^*(s,a) - \min_{(s,a) \in \mathcal{S} \times \mathcal{A}} Q^*(s,a)$. Then it can be shown that (see Appendix C)

$$\|\varepsilon_t\| \leq 2R_{\max} + \gamma \|Q^*\|_{\text{span}} := \kappa. \quad (6)$$

Moreover, it is easy to show that $\|\mathcal{G}_t(r_t)\| \leq \gamma \|r_t\|$, which follows from the contractive property of the empirical Bellman operator given the same next state. We shall say that $\mathcal{G}_t$ is quasi-contractive in the sense that the $\gamma$-contraction inequality only holds with respect to the origin $\mathbf{0}$.

**Inner SA:** We further characterize the dynamics of $\nu_t = Q_t^B - Q_t^A$ as an SA recursion (see Appendix C):

$$\nu_{t+1}(s,a) = (1 - \hat{\alpha}_t(s,a))\nu_t(s,a) + \hat{\alpha}_t(s,a)\left(\mathcal{H}_t(\nu_t)(s,a) + \mu_t(s,a)\right), \quad (7)$$

for all $t \geq 1$ where $\hat{\alpha}_t(s,a) := \begin{cases} \alpha_t, & \text{for synchronous version} \\ \alpha_t \tau_t(s,a), & \text{for asynchronous version} \end{cases}$. It has been shown in Xiong et al. (2020) that $\mathcal{H}_t$ is quasi-contractive satisfying $\|\mathcal{H}_t(\nu_t)\| \leq \frac{1+\gamma}{2}\|\nu_t\|$, and $\{\mu_t\}_{t \geq 1}$ is a martingale difference sequence with respect to the filtration $\mathcal{F}_t$ defined by $\mathcal{F}_1 = \{\emptyset, \Omega\}$ where $\Omega$ denotes the underlying probability space and for $t \geq 2$,

$$\mathcal{F}_t = \begin{cases} \sigma\left(\{s_k\}, \{R_{k-1}\}, \beta_{k-1}, 2 \leq k \leq t\right), & \text{for synchronous version,} \\ \sigma\left(s_k, a_k, R_{k-1}, \beta_{k-1}, 2 \leq k \leq t\right), & \text{for asynchronous version,} \end{cases} \quad (8)$$

where we note that for synchronous sampling $\{s_k\}$ and $\{R_{k-1}\}$ are the collections of sampled next states and the sampled rewards for each $(s,a)$-pair, respectively; while for asynchronous sampling, the pairs $\{(s_k, a_k, s_{k+1})\}_{k \geq 2}$ are consecutive sample transitions from one observed trajectory.

In the sequel, we will provide the finite-time convergence guarantee for (a)synchronous double Q-learning using the SA recursions described by (5) and (7).

### 3.2 FINITE-TIME BOUND FOR A GENERAL SA

In this subsection, we develop a convergence result for a general SA that will be applicable for both inner and outer SAs described in Section 3.1.

Consider the following general SA algorithm with the unique fixed point $\theta^* = 0$:

$$\theta_{t+1} = (1 - \alpha_t)\theta_t + \alpha_t\left(\mathcal{G}_t(\theta_t) + \varepsilon_t + \gamma\nu_t\right), \quad (9)$$

for all $t \geq 1$, where $\theta_t \in \mathbb{R}^n$ and we abuse the notation of a general learning rate $\alpha_t \in [0,1]$. Then we bound $\theta_t$ in the following proposition, the proof of which is provided in Appendix D.

**Proposition 1.** *Consider an SA given in (9). Suppose $\mathcal{G}_t$ is quasi-contractive with a constant parameter $\gamma$, that is, $\|\mathcal{G}_t(\theta_t)\| \leq \gamma \|\theta_t\|$ where $\gamma \in (0,1)$. Then for any learning rate $\alpha_t \in [0,1)$, the iterates $\{\theta_t\}$ satisfy*

$$\|\theta_t\| \leq \prod_{k=1}^{t-1}(1-(1-\gamma)\alpha_k)\|\theta_1\| + \gamma\alpha_{t-1}(\|W_{t-1}\| + \|\nu_{t-1}\|)$$

$$+ \gamma \sum_{k=1}^{t-2}\left\{\prod_{l=k+1}^{t-1}(1-(1-\gamma)\alpha_l)\right\}\alpha_k(\|W_k\| + \|\nu_k\|) + \|W_t\|, \tag{10}$$

*where the sequence $\{W_t\}$ is given by $W_{t+1} = (1-\alpha_t)W_t + \alpha_t\varepsilon_t$ with $W_1 = \mathbf{0}$.*

We note that an SA with a similar form to that in (9) has been analyzed in Wainwright (2019b), which additionally requires a monotonicity assumption. In contrast, our analysis does not require this assumption. Moreover, distinct from Wainwright (2019b), we treat the noise terms $\varepsilon_t$ and $\nu_t$ separately rather than bounding them together. This is because for double Q-learning, the noise term $\nu_t$ has its own dynamics which is significantly more complex than the i.i.d. noise $\varepsilon_t$. Bounding them as one noise term will yield more conservative results.

Note that the SA recursion (7) is a special case of (9) by setting $\nu_t = \mathbf{0}$. Therefore, Proposition 1 is readily applicable to both (5) and (7).

### 3.3 FINITE-TIME ANALYSIS OF SYNCHRONOUS DOUBLE Q-LEARNING

We apply the above bound for SA to synchronous double Q-learning and bound the error $\|r_t\| = \|Q_t^A - Q^*\|$. The first result is stated in the following theorem.

**Theorem 1.** *Fix $\gamma \in (0,1)$. Consider synchronous double Q-learning in (3) with a rescaled linear learning rate $\alpha_t = \frac{3}{3+(1-\gamma)t}, \forall t \geq 0$. Then the learning error $r_t = Q_t^A - Q^*$ satisfies*

$$\mathbb{E}\|r_{t+1}\| \leq \frac{3\|r_1\|}{(1-\gamma)t} + \frac{3\sqrt{3}\kappa\tilde{C}}{(1-\gamma)^{3/2}}\frac{1}{\sqrt{t}} + \frac{36\sqrt{3}V_{\max}\tilde{D}}{(1-\gamma)^{5/2}}\frac{1}{\sqrt{t}}, \tag{11}$$

*where $\tilde{C} := 6\sqrt{\ln 2D} + 3\sqrt{\pi}, \tilde{D} := 2\sqrt{\ln 2D} + \sqrt{\pi}$. and $\kappa$ is defined in (6) which is the uniform bound of $|\varepsilon_t|$.*

Theorem 1 provides the finite-time error bound for synchronous double Q-learning. To understand Theorem 1, the first term on the RHS (right hand side) of (11) shows that the initial error decays sub-linearly with respect to the number of iterations. The second term arises due to the fluctuation of the noise term $\varepsilon_t$, which involves the problem specific quantity $\kappa$. The last item arises due to the fluctuation of the noise term $\mu_t$ in the $\nu_t$-recursion (7), i.e., the difference between two Q-estimators.

**Corollary 1.** *The time complexity (i.e., the total number of iterations) to achieve an $\epsilon$-accurate optimal Q-function (i.e., $\mathbb{E}\|r_T\| \leq \epsilon$) is given by $T(\epsilon, \gamma, D) = \Omega\left(\frac{\ln D}{(1-\gamma)^7\epsilon^2}\right)$.*

*Proof.* The proof follows directly from Theorem 1 by noting that the middle term on the RHS of (11) scales as $\left(\frac{1}{1-\gamma}\right)^{\frac{5}{2}}$ since $\kappa = 2R_{\max} + \gamma\|Q^*\|_{\text{span}} \leq \frac{2R_{\max}}{1-\gamma} = V_{\max}$. $\qquad\square$

We next compare Corollary 1 with the time complexity of synchronous double Q-learning provided in Xiong et al. (2020), which is given by

$$T = \Omega\left(\left(\frac{1}{(1-\gamma)^6\epsilon^2}\ln\frac{D}{(1-\gamma)^7\epsilon^2}\right)^{\frac{1}{\omega}} + \left(\frac{1}{1-\gamma}\ln\frac{1}{(1-\gamma)^2\epsilon}\right)^{\frac{1}{1-\omega}}\right), \tag{12}$$

where $\omega \in (\frac{1}{3}, 1)$. For the $\epsilon$ dominated regime (with relatively small $\gamma$), the result in (12) clearly cannot achieve the order of $\frac{1}{\epsilon^2}$ and $\ln D$ as our result does. Further, its approaching to such an order ($\eta \to 0$ in Table 1) is also at an additional cost of an asymptotically large exponent on $\frac{1}{1-\gamma}$. For $1-\gamma$ dominated regime, the dependence on $1-\gamma$ can be optimized by taking $\omega = \frac{6}{7}$ in (12), compared to which our result achieves an improvement by a factor of $\mathcal{O}\left(\left(\ln\frac{1}{1-\gamma}\right)^7\right)$ (see Table 1).

### 3.4 FINITE-TIME ANALYSIS OF ASYNCHRONOUS DOUBLE Q-LEARNING

In this subsection, we provide the finite-time result for asynchronous double Q-learning. Differently from the synchronous version, at each iteration asynchronous double Q-learning only update one state-action pair of a randomly chosen Q-estimator. Thus the sampling strategy is important for the convergence analysis, for which we first make the following assumption.

**Assumption 1.** *The Markov chain induced by the stationary behavior policy $\pi$ is uniformly ergodic.*

This is a standard assumption under which Markov chain is most widely studied (Paulin et al., 2015). It was also assumed in (Qu & Wierman, 2020; Li et al., 2020) for the asynchronous samples in Q-learning. We further introduce the following standard notations (see for example Qu & Wierman (2020); Li et al. (2020)) that will be useful in the analysis.

First, we denote $\mu_\pi$ as the stationary distribution of the behavior policy over the state-action space $\mathcal{S} \times \mathcal{A}$ and denote $\mu_{\min} := \min_{(s,a)\in\mathcal{S}\times\mathcal{A}} \mu_\pi(s,a)$. It is easy to see that the smaller $\mu_{\min}$ is, the more iterations we need to visit all state-action pairs. Formally, we capture this probabilistic coverage by defining the following covering number:

$$L = \min\left\{t : \min_{(s_1,a_1)\in\mathcal{S}\times\mathcal{A}} \mathbb{P}(\mathcal{B}_t|(s_1,a_1)) \geq \frac{1}{2}\right\}, \tag{13}$$

where $\mathcal{B}_t$ denotes the event that all state-action pairs have been visited at least once in $t$ iterations.

In addition, the ergodicity assumption indicates that the distribution of samples will approach to the stationary distribution $\mu_\pi$ in a so-called mixing rate. We define the corresponding mixing time as

$$t_{\min} = \min\left\{t : \max_{(s_1,a_1)\in\mathcal{S}\times\mathcal{A}} d_{\mathrm{TV}}\left(P^t(\cdot|(s_1,a_1)),\mu_\pi\right) \leq \frac{1}{4}\right\}, \tag{14}$$

where $P^t(\cdot|(s_1,a_1))$ is the distribution of $(s_t,a_t)$ given the initial pair $(s_1,a_1)$, and $d_{\mathrm{TV}}(\mu,\nu)$ is the variation distance between two distributions $\mu,\nu$.

Next, we provide the first result for asynchronous double Q-learning in the following theorem whose proof is seen in Appendix H.

**Theorem 2.** *Fix $\gamma \in (0,1), \delta \in (0,1), \epsilon \in (0,\frac{1}{1-\gamma})$ and suppose that Assumption 1 holds. Consider asynchronous double Q-learning with a constant learning rate $\alpha_t = \alpha = \frac{c_1}{\ln\frac{DT}{\delta}}\min\left\{(1-\gamma)^6\epsilon^2,\frac{1}{t_{\min}}\right\}$ with some constant $c_1$. Then asynchronous double Q-learning learns an $\epsilon$-accurate optimum, i.e., $\left\|Q_t^A - Q^*\right\| \leq \epsilon$, with probability at least $1-\delta$ given the time complexity of*

$$T = \tilde{\Omega}\left(\left(\frac{1}{\mu_{\min}\epsilon^2(1-\gamma)^7} + \frac{t_{mix}}{\mu_{\min}(1-\gamma)}\right)\ln\frac{1}{\epsilon(1-\gamma)^2}\right),$$

*where $t_{mix}$ is defined in (14).*

The complexity in Theorem 2 is given in terms of the mixing time. To facilitate comparisons, we provide the following result in terms of the covering number.

**Theorem 3.** *Under the same conditions of Theorem 2, consider a constant learning rate $\alpha_t = \alpha = \frac{c_2}{\ln\frac{DT}{\delta}}\min\left\{(1-\gamma)^6\epsilon^2,1\right\}$ with some constant $c_2$. Then asynchronous double Q-learning can learn an $\epsilon$-accurate optimum, i.e., $\left\|Q_T^A - Q^*\right\| \leq \epsilon$, with probability at least $1-\delta$ given the time complexity of*

$$T = \tilde{\Omega}\left(\frac{L}{\epsilon^2(1-\gamma)^7}\ln\frac{1}{\epsilon(1-\gamma)^2}\right),$$

*where $L$ is defined in (13).*

We next compare Theorem 3 with the result obtained in Xiong et al. (2020). In Xiong et al. (2020), the authors provided the time complexity for asynchronous double Q-learning as

$$T = \Omega\left(\left(\frac{L^4}{(1-\gamma)^6\epsilon^2}\ln\frac{DL^4}{(1-\gamma)^7\epsilon^2}\right)^{\frac{1}{\omega}} + \left(\frac{L^2}{1-\gamma}\ln\frac{1}{(1-\gamma)^2\epsilon}\right)^{\frac{1}{1-\omega}}\right), \tag{15}$$

where $\omega \in (\frac{1}{3}, 1)$. It can be observed that our result improves that in (15) with respect to the order of all key parameters $\epsilon, D, 1 - \gamma, L$ (see Table 1). Specifically, the dependence on $L$ in (15) can be optimized by choosing $\omega = \frac{2}{3}$, upon which Theorem 3 improves by a factor of at least $L^5$.

### 3.5 Proof Sketch of Theorem 1

In order to provide the convergence bound for double Q-learning under the rescaled linear learning rate, we develop a different analysis approach from that in Xiong et al. (2020), the latter of which does not handle the rescaled linear learning rate. More specifically, in order to analyze a pair of nested SA recursions, we directly bound both the error dynamics of the outer SA between the Q-estimator and the global optimum and the error propagation between the two Q-estimators captured by the inner SA. Then we integrate the bound on the inner SA into that on the outer SA as a noise term, and establish the final convergence bound. This is a very different yet more direct approach than the techniques in Xiong et al. (2020) which constructs two complicated block-wisely decreasing bounds for the two SAs to characterize a block-wise convergence.

Our finite-time analysis for synchronous double Q-learning (i.e., Theorem 1) includes four steps.

**Step I: Bounding outer SA dynamics $\mathbb{E} \|r_t\|$ by inner SA dynamics $\mathbb{E} \|\nu_t\|$.** Here, $r_t := Q_t^A - Q^*$ captures the error dynamics between the Q-estimator and the global optimum, and $\nu_t := Q_t^A - Q_t^B$ captures the error propagation between the two Q-estimators. We apply Proposition 1 to the error dynamics (5) of $r_t$, take the expectation, and apply the learning rate inequality (24) to obtain

$$\mathbb{E} \|r_t\| \leq \alpha_{t-1} \|r_1\| + \frac{\gamma}{2} \alpha_{t-1} \sum_{k=1}^{t-1} (\mathbb{E} \|W_k\| + \mathbb{E} \|\nu_k\|) + \mathbb{E} \|W_t\|, \tag{16}$$

where $W_{t+1} = (1 - \tilde{\alpha}_t)W_t + \tilde{\alpha}_t \varepsilon_t$, with initialization $W_1 = \mathbf{0}$.

**Step II: Bounding $\mathbb{E} \|W_t\|$.** We first construct a $\mathcal{F}_t$-martingale sequence $\{\tilde{W}_i\}_{1 \leq i \leq t+1}$ with $\tilde{W}_{t+1} = W_{t+1}$ and $\tilde{W}_1 = \mathbf{0}$. Next, we bound the squared difference sequence $(\tilde{W}_{i+1} - \tilde{W}_i)^2$ by $4V_{\max}^2 \alpha_t^N / \alpha_i^{N-2}$, for $1 \leq i \leq t$, where $N$ is defined in (26). Then we apply the Azuma-Hoeffding inequality (see Lemma 5) to $\{\tilde{W}_i\}_{1 \leq i \leq t+1}$ and further use Lemma 6 to obtain the bound on $\mathbb{E} \|W_t\|$ in Proposition 2 which is given by

$$\mathbb{E} \|W_{t+1}\| \leq \kappa \tilde{C} \sqrt{\alpha_t}, \tag{17}$$

where $\tilde{C} = 6\sqrt{\ln 2D} + 3\sqrt{\pi}$ and $\kappa$ is defined in (6).

**Step III: Bounding inner SA dynamics $\mathbb{E} \|\nu_t\|$.** Similarly to Step I, we apply Proposition 1 to the $\nu_t$-recursion (7), take the expectation, and apply the learning rate inequality (24) to obtain

$$\mathbb{E} \|\nu_t\| \leq \alpha_{t-1} \|\nu_1\| + \frac{1+\gamma}{2} \alpha_{t-1} \sum_{k=2}^{t-1} \mathbb{E} \|M_k\| + \mathbb{E} \|M_t\|, \tag{18}$$

where $M_{t+1} = (1 - \alpha_t)M_t + \alpha_t \mu_t$, with initialization $M_1 = \mathbf{0}$. Using a similar idea to Step II, we obtain the bound on $\mathbb{E} \|M_t\|$ in Proposition 3. Finally, we substitute the bound of $\mathbb{E} \|M_t\|$ back in (18) and use the fact $\|\nu_1\| = 0$ to obtain

$$\mathbb{E} \|\nu_t\| \leq \frac{6V_{\max}\tilde{D}}{1 - \gamma} \sqrt{\alpha_{t-1}}, \quad \text{with } \tilde{D} = 2\sqrt{\ln 2D} + \sqrt{\pi}. \tag{19}$$

**Step IV: Deriving finite-time bound.** Substituting (17) and (19) into (16) yields (11).

## 4 Conclusion

In this paper, we derived sharper finite-time bounds for double Q-learning with both synchronous sampling and Makovian asynchronous sampling. To achieve this, we developed a different approach to bound two nested stochastic approximation recursions. An important yet challenging future topic is the convergence guarantee for double Q-learning with function approximation. In addition to the lack of the contraction property of the Bellman operator in the function approximation setting, it is likely that neither of the two Q-estimators converges, or they do not converge to the same point even if they both converge. Characterizing the conditions under which double Q-learning with function approximation converges is still an open problem.

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
