# OpenReview forum: "Double Q-learning: New Analysis and Sharper Finite-time Bound"
_ICLR.cc/2021/Conference — Reject_

### Official Review · AnonReviewer4 · 2020-10-28
**significance of contributions, discussions on prior works**

**Rating:** 6
**Confidence:** 3

**Review:**

This paper provides a sharper analysis for the finite time convergence rate of the double Q learning algorithm. The authors provides bounds for the synchronous and asynchronous settings and uses a more refined learning rate of $a/(b+t)$. It is shown that with such step size rule, a sharper convergence rate than (Xiong et al., 2020) can be obtained.

The advantages of this paper are that (i) the authors analyzed a popular algorithm with reasonable assumptions; (ii) particularly, the analysis which is achieved by studying a nested stochastic approximation scheme with the Azuma-Hoeffding inequality is quite easy to follow. The reviewer believes that the analysis may generate new insights for the future work in related domains.

There are some outstanding concerns/comments as follows:

- Significance of contribution

As the authors mentioned, the double Q learning algorithm has been analyzed in the prior works such as (Xiong et al., 2020). While the obtained rates by the authors are sharper, in the current presentation, the reviewer finds the result relatively incremental. It may be useful to better highlight the difference between the analysis approach in this paper and (Xiong et al., 2020).

Moreover, though the double Q learning algorithm is different from the standard Q learning, it also seems that the sharper analysis done in this paper has a worse dependence on $1-\gamma$ compared to (Qu & Wierman, 2020).

- Relation to Prior Works

As claimed by the authors, one of the major innovations in this work is to deploy a rescaled step size of the form $a/(b+ct)$. However, it should be noted using such rescaled step size is also common in the stochastic approximation analysis, e.g.,

Bhandari et al., A Finite Time Analysis of Temporal Difference Learning With Linear Function Approximation, COLT, 2018.

Also, the idea of analyzing nested stochastic approximation can be found in a few recent works on analyzing 2 timescale stochastic approximation, e.g.,

T. Doan, Finite-time analysis and restarting scheme for linear two-time-scale stochastic approximation, arXiv/1912.10583.

Dalal et al., A Tale of Two-Timescale Reinforcement Learning with the Tightest Finite-Time Bound, AAAI 2020.

Kaledin et al., Finite time analysis of linear two-timescale stochastic approximation with Markovian noise, COLT 2020.

It would put the paper in a better position if the authors could emphasize on how the analysis is related to the above mentioned works.

- Async. Q learning

In (4), the authors mentioned that the state-action pair change over $t$ without a further discussion on how $a_t,s_t$ are generated. From the analysis in Section 3, it seems that the analysis can tackle the general cases satisfying assumption 1 (including the ergodicity assumption ones). It maybe beneficial to supplement the discussions in Section 2 with a few concrete examples.

Lastly, the current bound developed by the paper has a dependence of $L^3$ (cf. Assumption 1), where $L \geq D$, which seems to be quite high when the state/action space is large. Particularly, it is also worse than the $L^2$ bound analyzed in (Qu and Wierman, 2020) - of course, the latter paper analyzed a different algorithm, but it would again put the paper in a better position if the authors could discuss the differences in these convergence rates.

* Minor point: the reviewer wonders that in Table 1, the convergence bound for "This Work" is shown for both $\omega = 1-\eta \rightarrow 1$ and $\omega = 6/7$. From my understanding, the $\omega$ exponent is always $1$ for this work.

Post-rebuttal: I am satisfied with the authors' response and decided to keep my score.

---

> ### Author Response · Authors · 2020-11-19
> **Thanks for the comments! We have addressed all the concerns and imrpoved the dependence on $L$.**
>
> We thank the reviewer for the very helpful comments, which significantly help to improve the quality of the paper.
>
> Q1: Significance of contribution: It may be useful to better highlight the difference between the analysis approach in this paper and (Xiong et al., 2020).
>
> A1: First, this paper improves the dependence on all the key parameters ($\epsilon$, $1-\gamma$, $|S||A|$) in the time complexity. Second, as for the analysis approach, (Xiong et al., 2020) constructs complicated block-wise upper bounds to show the convergence, and such an approach is not applicable to handle the linear learning rate or constant learning rate. Our approach here is very different and directly analyzes the evolution of the gap between two SAs. Section 3.5 provides a more detailed explanation about our analysis framework.
>
> Q2: Though the double Q learning algorithm is different from the standard Q learning, it also seems that the sharper analysis done in this paper has a worse dependence on $1-\gamma$ compared to (Qu & Wierman, 2020).
>
> A2: First, it has been demonstrated that double Q-learning is empirically useful, particularly overcoming the overestimation, rather than being designed to accelerate Q-learning. Intuitively, it is arguably reasonable to expect that double Q-learning by nature may have an inferior rate than Q-learning, because the design of double Q-learning does yield more conservative updates than Q-learning. Hence, this study aims at providing a refined analysis for double Q-learning instead of showing a faster convergence rate than vanilla Q-learning. We also provide an inclusive understanding on how the convergence rate of double Q-learning compares with Q-learning, which is certainly meaningful.
>
> Q3: As claimed by the authors, one of the major innovations in this work is to deploy a rescaled step size of the form . However, it should be noted using such rescaled step size is also common in the stochastic approximation analysis. It would put the paper in a better position if the authors could emphasize on how the analysis is related to the above mentioned works.
>
> A3: Many thanks for the suggestion! The listed works by the reviewer mainly deal with a single SA using rescaled linear rates. The two-timescale SA uses an auxiliary parameter for a better estimation of the update in the main parameter. Essentially, these studies analyze the convergence of a single SA sequence. In contrast, double Q-learning involves two SAs evolving dynamically in a randomly switching manner. Hence, the main analysis challenge lies in how to deal with such interconnected SAs and analyze their convergence. Section 3.5 provides a more detailed explanation of our analysis framework. In the revision, we have added the suggested references and the corresponding discussion in Appendix A.
>
> Q4: In (4), the authors mentioned that the state-action pair changes over without a further discussion on how $(s_t,a_t)$ are generated. From the analysis in Section 3, it seems that the analysis can tackle the general cases satisfying assumption 1 (including the ergodicity assumption ones). It may be beneficial to supplement the discussions in Section 2 with a few concrete examples.
> Lastly, the current bound developed by the paper has a dependence of $L^3$ (cf. Assumption 1), where $L\geq D$, which seems to be quite high when the state/action space is large. Particularly, it is also worse than the $L^2$ bound analyzed in (Qu and Wierman, 2020) - of course, the latter paper analyzed a different algorithm, but it would again put the paper in a better position if the authors could discuss the differences in these convergence rates.
>
> A4: Regarding Assumption 1, based on the suggestion of Reviewer #1, in the revision, we have changed our assumption to a more practical ergodicity one which has been used in [Li et. al., 2020, Qu et. al. 2020] and derived new results in Section 3.4. In particular, we managed to improve the dependence from L^6 to L (see theorems in Section 3.4 and corresponding proofs in Appendices H-K).
>
> Q5: Minor point: the reviewer wonders that in Table 1, the convergence bound for "This Work" is shown for both $\omega=1-\eta\rightarrow 1$ and $\omega=6/7$. From my understanding, the $\omega$ exponent is always 1 for this work.
>
> A5: Yes. Linear learning rate requires $\omega=1$. However, the results in [Xiong et. al., 2020] cannot cover this case.

---

### Official Review · AnonReviewer1 · 2020-10-30

**Rating:** 4
**Confidence:** 4

**Review:**

This paper studies the convergence rate of double Q learning under the tabular setting. Both the synchronous and asynchronous double Q learning algorithms are studied and analyzed. The technical novelty of this paper seems limited -- possibly a direct combination of [Wainwright 2019a] and [Xiong et al 2020]. Moreover, the scope of this work seems also limited -- it only consider the tabular version of double Q-learning, with either a generative model (synchronous) or i.i.d. sampling (asynchronous) sampling models, which is never used in practice. This seems to makes this work only appealing to theorists. However, in terms of the theory, the results for double Q learning is pessimist -- the rate is much worse than standard Q learning. However, practical implementations of double Q learning has demonstrated its advantage in terms of correcting the over-estimation bias. Without numerical results or additional theory, such a gap makes one ponder whether the rates in this paper are tight.


Detailed comments:

1. This paper seems a theory paper, with the focus on understanding the convergence rate of double Q learning under the tabular case. This paper provides a unified analysis for both the synchronous and asynchronous settings using stochastic approximation (Proposition 1). However, seen from the proof, this proposition is a modification of Theorem 1 in [Wainwright 2019a]. Based on this stochastic approximation result, showing the convergence rate for double Q learning by analyzing the evolution of outer and inner stochastic approximations follows from [Xiong et al 2020]. Thus, the technical novelty of this paper is limited.

2. The synchronous setting seems too restrictive because one needs to update the Q value of every state and action at each iteration. In other words, this setting essentially assumes a version of the generative model.
A more interesting setting is the asynchronous setting, where one updates each state-action at each iteration. However, this paper makes the assumption that the samples are i.i.d. and each state-action appears with probability at least 1/L. This is very restrictive and essentially reduces the sampling model to the generative model, as one can first estimate the model and solve a planning problem. This approach is also known as minimax optimal. A more practical assumption is to sample a trajectory according to a behavioural policy, as is done in [Li et al. 2020] (Sample Complexity of Asynchronous Q-Learning:
Sharper Analysis and Variance Reduction) for asynchronous Q Learning.

3. Compared with the results for standard Q Learning, e.g., [Li et al. 2020], the rates obtained here are inferior. This seems somewhat disappointing because there seems no advantage of using double Q Learning -- the algorithm is more complicated and the rate is worse.

4. It would be interesting to see whether (1) variance reduction and (2) constant stepsize can be used to sharpen the rates.

5. I appreciate the efforts of the authors in comparing with the related work details. It would be nice to also compare with papers on standard Q-learning and other methods in the tabular settings.

6. In the abstract, why is the run-time complexity given by the "big Omega" notation, which means that the time-complexity is larger than the quantity provided in terms of the order. It seems that the complexity should be given in "big O" notation.




Missing references:

**Tabular Q-learning**:

Is Q-Learning Provably Efficient?

Q-learning with Logarithmic Regret

Q-learning with UCB Exploration is Sample Efficient for Infinite-Horizon MDP

Periodic Q-Learning

**Q-Learning with function approximation**:

Sample-Optimal Parametric Q-Learning Using Linearly Additive Features

Feature-based q-learning for two-player stochastic games

Adaptive Discretization for Episodic Reinforcement Learning in Metric Spaces

Q-learning with Nearest Neighbors

An Analysis of Reinforcement Learning with Function Approximation

A Theoretical Analysis of Deep Q-Learning

Finite-Sample Analysis of Nonlinear Stochastic Approximation with Applications in Reinforcement Learning


**Tabular RL**:

Variance reduced value iteration and faster
algorithms for solving Markov decision processes

Primal-Dual π Learning: Sample Complexity and Sublinear Run Time for Ergodic Markov Decision Problems

Model-Based Reinforcement Learning with a Generative Model is Minimax Optimal

Efficiently Solving MDPs with Stochastic Mirror Descent

Near-Optimal Time and Sample Complexities for Solving Discounted Markov Decision Process with a Generative Model

Variance Reduced Value Iteration and Faster Algorithms for Solving Markov Decision Processes

Minimax Optimal Reinforcement Learning for Discounted MDPs

---

> ### Author Response · Authors · 2020-11-19
> **Thanks for the comments! We have addressed all the concerns, changed the assumption as suggested and improved the dependence on $L$.**
>
> We thank the reviewer for the very helpful comments, which significantly help to improve the quality of the paper.
>
> Q1: This paper provides a unified analysis for both the synchronous and asynchronous settings using stochastic approximation (Proposition 1). However, seen from the proof, this proposition is a modification of Theorem 1 in [Wainwright 2019a]. Based on this stochastic approximation result, showing the convergence rate for double Q learning by analyzing the evolution of outer and inner stochastic approximations follows from [Xiong et al 2020]. Thus, the technical novelty of this paper is limited.
>
> A1: In fact, this paper uses a completely different analysis framework from [Xiong et al 2020] to handle the coupling of two inter-connected SAs, because [Xiong et al 2020]’s technique is not applicable to rescaled linear learning rates. This is also the reason why we have an improved sample complexity bound.
>
> Q2: A more interesting setting is the asynchronous setting, where one updates each state-action at each iteration. However, this paper makes the assumption that the samples are i.i.d. and each state-action appears with probability at least $1/L$. This is very restrictive and essentially reduces the sampling model to the generative model, as one can first estimate the model and solve a planning problem. This approach is also known as minimax optimal. A more practical assumption is to sample a trajectory according to a behavioural policy, as is done in [Li et al. 2020] (Sample Complexity of Asynchronous Q-Learning: Sharper Analysis and Variance Reduction) for asynchronous Q Learning.
>
> A2: Thank you very much for the comments! We fully agree. We have changed our assumption to the ergodicity assumption in the revision (see Section 3.4). Furthermore, we were also able to further improve the sample complexity for the asynchronous setting from $L^6$ to $L$ (see theorems in Section 3.4 and the corresponding proofs in Appendices H-K).
>
> Q3: Compared with the results for standard Q Learning, e.g., [Li et al. 2020], the rates obtained here are inferior. This seems somewhat disappointing because there seems no advantage of using double Q Learning -- the algorithm is more complicated and the rate is worse.
>
> A3: First, it has been demonstrated that double Q-learning is empirically useful, particularly overcoming the overestimation, rather than being designed to accelerate Q-learning. Intuitively, it is arguably reasonable to expect that double Q-learning by nature may have an inferior rate than Q-learning, because the design of double Q-learning does yield more conservative updates than Q-learning. Hence, this study does not aim at showing a faster convergence rate than Q-learning, which may not hold at the first place, but rather provides an inclusive understanding on how the convergence rate of double Q-learning compares with Q-learning, which is certainly meaningful.
>
> Q4: It would be interesting to see whether (1) variance reduction and (2) constant stepsize can be used to sharpen the rates.
>
> A4: Many thanks for the suggestion!
> i) For constant stepsize, we managed to work out the analysis and have added the new result in Section 3.4 with the corresponding proofs in Appendices H-K, which can indeed improve the dependence on $L$. We also emphasize that the extension from asynchronous Q-learning with constant learning rates [Li et. al., 2020] to double Q-learning is highly non-trivial. The key challenges are to deal with the switching random variable ($\beta_t$ in the paper) and the inter-connection between two Q-estimators ($\nu_t$ in the paper), which are unique in the analysis of double Q-learning and need considerable effort.
> ii) For variance reduction, we are still working along this direction. So far, we found these extensions may be natural for Q-learning, but we encountered quite a bit of technical challenges in double Q-learning due to the inter-connected SAs.
>
> Q5: I appreciate the efforts of the authors in comparing with the related work details. It would be nice to also compare with papers on standard Q-learning and other methods in the tabular settings.
>
> A5: Thanks for the suggestion. We have added a section (Appendix A) to discuss more related work (including those suggested by the reviewer) in the supplementary material.
>
> Q6: In the abstract, why is the run-time complexity given by the "big Omega" notation, which means that the time-complexity is larger than the quantity provided in terms of the order. It seems that the complexity should be given in "big O" notation.
>
> A6: We use “big Omega” to indicate that the complexity should be at least at such a level to achieve $\epsilon$-accuracy.

---

### Official Review · AnonReviewer2 · 2020-10-31
**Review for Double Q-learning: New Analysis and Sharper Finite-time Bound**

**Rating:** 6
**Confidence:** 3

**Review:**

This paper provides a new theoretical analysis of double Q-learning in the tabular case. The analysis improves over previous result of Xiong et al. which assumes polynomial learning rate. This paper considers rescaled linear learning rate and the sample complexity has better dependency on 1/eps. The improvement comes from a better characterization of the error dynamics.

Overall, this paper is well-written. The authors provide a careful comparison with previous results, and also provide a technical overview to highlight the new ideas in the analysis. The authors also provide a proof sketch in Section 3.5.

My main concern is whether the results are interesting enough to the ICLR community. Certainly, understanding the theoretical guarantee double Q-learning is an important topic, and this paper provides a nice improvement over the previous analysis. However, I am not sure if the improvement is significant enough compared to the previous result. Moreover, the bounds provided in this paper is still far from being practical (in the synchronous setting, all state-action pairs are visited in each iteration, and in the asynchronous setting the sample complexity is at least (|S||A|)^6. Given this, my commendation would merely be a weak acceptance.

***Post Rebuttal***
I appreciate the authors's improved analysis. After reading the new version it is unclear to me whether the results are interesting enough to the ICLR community, and thus I would like to keep my original score.

---

> ### Author Response · Authors · 2020-11-19
> **Thanks for the comments! We have improved the sample complexity to a more practical one.**
>
> We thank the reviewer for the very helpful comments, which significantly help to improve the quality of the paper.
>
> Q1: I am not sure if the improvement is significant enough compared to the previous result.
>
> A1: Our result of orderwise improvement is not only theoretically meaningful, but also suggests that the rescale linear learning rate should yield better sample efficiency in practice than the polynomial learning rate adopted by the previous study. In Appendix B, we have also added a numerical experiment, which verifies such an insight from theory.
>
> Q2: Moreover, the bounds provided in this paper are still far from being practical (in the synchronous setting, all state-action pairs are visited in each iteration, and in the asynchronous setting the sample complexity is at least $(|S||A|)^6$).
>
> A2: We thank the reviewer very much for this comment! We have managed to improve the order for the asynchronous setting from $L^6$ to $L$ (where $L$ is at least $|S||A|$), by more refined bounding techniques and using a constant learning rate. In the revision, Table 1 reflects our new bound and Section 3.4 includes all the new results which are highlighted in blue colored texts. The corresponding new proofs are provided in Appendices H-K.

---

### Official Review · AnonReviewer3 · 2020-11-03
**The paper presents some new interesting ideas, but it is unclear how meaningful  the new bounds are.**

**Rating:** 5
**Confidence:** 3

**Review:**

This paper refines the existing finite-sample bound of double Q-learning. This paper considers the rescaled linear schedule of the learning rate and claims that the sample complexity bounds are improved in the sense the dependence on all main parameters (epsilon, 1-gamma, L, D) have been improved.


Originality: The paper follows the work in Xiong et.al 2020, but has made some non-trivial improvements. The nested SA representation of double Q-learning is interesting and the proving techniques seem new.

Quality: The technical quality of this paper looks reasonably good to me although I have only roughly checked the proofs. The high level idea of the proof makes sense to me. I haven't found any flaws in the proof sketches.

Clarity: Overall the paper is well written.

Significance: I am not sure whether the significance level of this submission meets the standard of ICLR or not. The main reason is that the bounds in this paper are on the expectation of L1 norm of the iteration error. The bounds in the Xiong et.a. (2020) are high probability bounds. Is it fair to make comparisons between these two? The L1 bound is  weaker than the mean square bounds and I am not fully convinced how meaningful such results are. In addition, the implications of the proposed theory on algorithm design have not been fully verified on numerical examples.

Pros:
1. The linear learning rate schedule is considered.
2. The nested SA representation is interesting and the proof technique looks new.


Cons:
1. The bounds on the iteration errors are in the L1 sense and it is unclear how meaningful these bounds are. Is the comparison with Xiong et.al (2020) really fair?
2. Does the theory in this paper lead to any new insight for design and tuning of double Q-learning? Any numerical justifications for these new insights?

---

> ### Author Response · Authors · 2020-11-19
> **Thanks for the comments! We have addressed the concerns and added an illustrative numerical example.**
>
> We thank the reviewer for the very helpful comments, which significantly help to improve the quality of the paper.
>
> Q1: The bounds on the iteration errors are in the L1 sense and it is unclear how meaningful these bounds are. Is the comparison with Xiong et.al (2020) really fair?
>
> A1: L1 norm has been extensively used in the theoretical studies of Q-learning. It is well known that the expectation of the maximum norm bound is equivalent to the high probability bound in [Xiong et.al (2020)] by the Chernoff-type inequalities. This has been commented in [page 10, footnote, Wainwright 2019, https://arxiv.org/abs/1905.06265]. Thus, the result here is comparable to Xiong et.al (2020).
>
>
> Q2: Does the theory in this paper lead to any new insight for design and tuning of double Q-learning? Any numerical justifications for these new insights?
>
> A2: Great question! This is exactly the contribution of this paper beyond Xiong et.al (2020). Our theory shows that using the rescaled linear learning rate provably yields order-level better sample efficiency bound compared to the polynomial learning rate in Xiong et.al (2020), and hence suggests that the practical implementation of double Q-learning should adopt such a learning rate.
> We have also added a numerical experiment in Appendix B, which verifies that the rescaled linear learning rate indeed improves the convergence of synchronous double Q-learning over the polynomial learning rate.

---

### Decision · Program_Chairs · 2021-01-07
**Final Decision**

**Decision:**

Reject

**Comment:**

The paper provides an improved analysis of the finite time convergence rate of double Q-learning under more reasonable step size rules, comparing to previous work by Xiong et al., 2020.  Understanding the convergence behavior of double Q-learning is an obviously interesting theoretical topic and all reviewers appreciate the authors’ improved analysis.

 Several reviewers questioned the sample complexity in terms of the dependence on L (thus |S||A|); In the latest revision, the authors claimed they now refined the dependence from O(L^6) to O(L). This major change is yet to be further reviewed since the authors did not leave any clue on why/how such an improvement was attained.
Another outstanding concern relates to the theoretical comparison of the rates between double Q-learning and Q-learning, which remains clueless. It’s unclear whether the bound in this paper is sharp enough and whether/when double Q-learning is provably inferior than Q-learning.

Therefore, I am not recommending acceptance at this time, though I encourage the authors to resubmit with a more conclusive theoretical analysis.